# A New Type of Shape-Invariant Beams with Structured Coherence: Laguerre-Christoffel-Darboux Beams

**Rosario Martínez-Herrero** [1,†] , **Massimo Santarsiero** [2,*,†] , **Gemma Piquero** [1,†]
**and Juan Carlos González de Sande** [3,†]

1 Departamento de Óptica, Universidad Complutense de Madrid, Ciudad Universitaria, 28040 Madrid, Spain; r.m-h@fis.ucm.es (R.M.-H.); piquero@ucm.es (G.P.)
2 Dipartimento di Ingegneria, Università Roma Tre, Via V. Volterra 62, 00146 Rome, Italy
3 ETSIS de Telecomunicación, Campus Sur, Universidad Politécnica de Madrid, 28031 Madrid, Spain; juancarlos.gonzalez@upm.es
* Correspondence: msantarsiero@uniroma3.it
† These authors contributed equally to this work.

**Abstract:** A new class of sources presenting structured coherence properties is introduced and analyzed. They are obtained as the incoherent superposition of coherent Laguerre-Gaussian modes with suitable coefficients. This ensures that the shape of the intensity profile and the spatial coherence features of the propagated beams are invariant during paraxial approximation. A simple analytical expression is obtained for the cross-spectral density of the sources of this class, regardless of the number of superposed modes. Properties of these sources are analyzed and described by several examples.

**Keywords:** coherence; structured light; propagation; shape-invariant fields





## 1. Introduction

The spatial coherence properties of a scalar planar source up to the second order are taken into account by its cross-spectral density function, or CSD, which gives the correlation between the field values at two distinct points of the plane at any temporal frequency. On the CSD of a source depend, in particular, also the intensity and coherence properties of the field it radiates [1].

The research on possible forms of CSD that give rise to fields with peculiar propagation properties has been going on continuously since the first seminal papers of the late 1970s [2–5]. One of the difficulties one encounters in devising new forms of CSDs is due to the fact that the latter are not generic functions of two points in space. In fact, necessary and sufficient condition for them to represent a valid CSD is that they are kernels of nonnegative definite integral hermitian operators [1]. When this happens, the CSD is said to be genuine, or *bona-fide*. If such requirement is not met, the function cannot represent the CSD of a possible physical source. In general, it is not easy to check the nonnegativity of an integral kernel, even though genuineness criteria have been introduced to this aim [6,7].

A different way, but equivalent to the above one, to envisage new forms of CSD consists in starting from the modal expansion of the CSD. In fact, on exploiting a fundamental result of coherence theory, any bona-fide CSD can be written as the (possibly infinite) sum of terms, each of them corresponding to the CSD of a perfectly coherent light beam (a mode of the source) [1,7–9]. Each term is weighted by a non-negative coefficient (the eigenvalue). In physical terms, this means that any partially coherent source can be thought of as the superposition of an ensemble of mutually uncorrelated, perfectly coherent fields. The eigenvalues then play the role of the powers of the various modes in the superposition.

On summing the CSDs of a number of coherent fields, of course, the resulting CSD is genuine but, on the down side, the obtained result is rarely in closed form, that is, it is generally not given in terms of simple mathematical functions. On the contrary, this

is a fundamental requirement in order for the source features, as well as the ones of the propagated fields, to be deduced from the analytical properties of the involved functions. Moreover, when the number of the superimposed modes is very large, the availability of a closed formula for the CSD turns out to be useful even from a computational point of view.

As an example, the modes of the so-called Gaussian Schell-model (GSM) sources [10,11] that will be recalled later on, are Hermite–Gaussian (in rectangular coordinates) or Laguerre–Gaussian (LG for short, in 2D polar coordinates) functions, i.e., the ones that represent the steady states of a quantum harmonic oscillator [12]. The corresponding eigenvalues decrease geometrically with the mode order. The final result of the sum of infinite terms is a CSD whose intensity and coherence profiles have simple Gaussian shapes.

In the present paper we introduce a new class of partially coherent sources that are obtained as the superposition of a finite number of LG modes. Whatever the number of modes involved, the CSDs of this class can be always be written in a simple closed form, whose expression depends on the number of the modes, but also on a second parameter, namely, the angular index (or topological charge) of the LG functions. In such a way, the structuring characteristics of the source (concerning both its intensity and degree of coherence) can be varied to a great extent on changing the above parameters. Due to the mathematical expression giving rise to the resulting closed form, sources of this class will be called *Laguerre-Christoffel-Darboux* (LCD) sources. Analogous sources have been recently introduced for the onedimensional case [13]. It will be seen that in some cases the intensity of the resulting source has a donut-like profile and it is practically uniform and vanishing in the inner and outer regions of the donut.

A significant feature of LCD sources, due to the propagation properties of the underlying modes, is that the fields radiated from them keep their shape unchanged during paraxial propagation at any distance from the source, except for a transverse scaling factor and a spherical curvature term. Such invariance concerns the whole CSD of the field (and, therefore, both the intensity profile and the coherence properties) and we will say that their CSDs are *shape-invariant*.

Another reason of interest for sources having modes of the LG type comes from the fact that such functions present wave-front singularities, known as optical vortices, and carry optical angular momentum (OAM). The presence of structures of this kind in coherent and partially coherent light beams is nowadays the subject of many research works (see for example [14–21] and references therein).

The paper is structured as follows. In Section 2 the concepts that are at the basis of the modal theory of coherence are recalled, as well as some known examples of partially coherent source obtained as superpositions of LG modes. In Section 3 the LCD sources are introduced, and their main features are analyzed in details. Finally, the main results of this work are summarized in Section 4.

## 2. Preliminaries

Partially coherent planar sources can be appropriately described by their CSD, $W(\boldsymbol{r}_1, \boldsymbol{r}_2, 0)$ [1], which gives the second-order correlations of the field at two different points $\boldsymbol{r}_1$ and $\boldsymbol{r}_2$ of the source plane. Here and in the following we omit the explicit dependence of $W$ on the temporal frequency. From such CSD function, the intensity can be derived by evaluating it at coincident points, that is,

$$I(\boldsymbol{r}, 0) = W(\boldsymbol{r}, \boldsymbol{r}, 0) . \tag{1}$$

A normalized version of the CSD function is the complex degree of coherence, which is defined as

$$\gamma(\boldsymbol{r}_1, \boldsymbol{r}_2, 0) = \frac{W(\boldsymbol{r}_1, \boldsymbol{r}_2, 0)}{\sqrt{I(\boldsymbol{r}_1, 0) I(\boldsymbol{r}_2, 0)}} , \tag{2}$$

whose absolute value ranges form 0 to 1, the two extremes denoting (spatial) complete incoherence and perfect coherence, respectively.

According to Mercer's theorem [1,22], any valid CSD function can be expanded as a series of the following type

$$W(\boldsymbol{r}_1,\boldsymbol{r}_2,0) = \sum_{n,m} \lambda_{nm}\, \Phi^*_{nm}(\boldsymbol{r}_1,0)\Phi_{nm}(\boldsymbol{r}_2,0),$$

(3)

where $n$ and $m$ are indices denoting the eigenfunctions $\Phi_{nm}(\boldsymbol{r},0)$ of the homogeneous Fredholm integral equation of the second kind whose kernel is $W(\boldsymbol{r}_1,\boldsymbol{r}_2,0)$, and $\lambda_{nm}$ the corresponding eigenvalues [1]. Due to the non-negative definiteness of $W$, all the eigenvalues are nonnegative. Using this modal expansion, the intensity distribution across the source can be expressed as

$$I(\boldsymbol{r},0) = \sum_{n,m} \lambda_{nm}|\Phi_{nm}(\boldsymbol{r},0)|^2.$$

(4)

The use of the modes allows one to evaluate the CSD of the field propagated from the source in an easy way, as the sum of the CSDs of the propagated modes. In fact we have

$$W(\boldsymbol{r}_1,\boldsymbol{r}_2,z) = \sum_{n,m} \lambda_{nm}\, \Phi^*_{nm}(\boldsymbol{r}_1,z)\Phi_{nm}(\boldsymbol{r}_2,z)$$

(5)

and, consequently,

$$I(\boldsymbol{r},z) = \sum_{n,m} \lambda_{nm}|\Phi_{nm}(\boldsymbol{r},z)|^2.$$

(6)

### 2.1. Shape Invariant Partially Coherent Fields: LG Modes

A particular set of orthonormal mode is formed by the LG functions, defined as

$$\Phi_{nm}(\boldsymbol{r},0) = \frac{1}{w_0}\sqrt{\frac{2n!}{\pi(n+|m|)!}}\, e^{-r^2/w_0^2}\, e^{im\varphi}\left(\frac{\sqrt{2}\,r}{w_0}\right)^{|m|} L_n^{|m|}\left(\frac{2r^2}{w_0^2}\right),$$

(7)

where $n = 0,1,\dots,\infty$, $m = 0,\pm 1,\dots,\pm\infty$, $\boldsymbol{r} = (r,\varphi)$, and $L_n^m$ are the associated Laguerre polynomials [23,24]. The parameter $w_0$ (the spot size) specifies the transverse width of the functions.

By substituting Equation (7) into the Mercer's expansion of Equation (3), a bona-fide CSD is obtained as

$$W(\boldsymbol{r}_1,\boldsymbol{r}_2,0) \quad = \frac{2}{\pi\, w_0^2}\, e^{-(r_1^2+r_2^2)/w_0^2} \times$$

(8)

$$\sum_{n,m} \lambda_{nm}\, \frac{n!}{(n+|m|)!}\left(\frac{2r_1 r_2}{w_0^2}\right)^{|m|} L_n^{|m|}\left(\frac{2r_1^2}{w_0^2}\right) L_n^{|m|}\left(\frac{2r_2^2}{w_0^2}\right) e^{-im(\varphi_1-\varphi_2)}.$$

Assuming paraxial approximation, the free-space propagated LG modes can be calculated as [23]

$$\Phi_{nm}(\boldsymbol{r},z) = \quad \frac{1}{w_z}\sqrt{\frac{2n!}{\pi(n+|m|)!}}\, e^{-r^2/w_z^2}\, e^{ikr^2/(2R_z)}\, e^{im\varphi} \times$$

(9)

$$e^{i[kz-(2n+|m|+1)\alpha_z]}\left(\frac{\sqrt{2}\,r}{w_z}\right)^{|m|} L_n^{|m|}\left(\frac{2r^2}{w_z^2}\right),$$

where the parameters $w_z$, $R_z$ and $\alpha_z$ are

$$w_z = w_0\sqrt{1+\left(\frac{z}{z_R}\right)^2}\;;\quad R_z = z\left[1+\left(\frac{z_R}{z}\right)^2\right];\quad \alpha_z = \arctan\left(\frac{z}{z_R}\right),$$

(10)

$z_R = kw_0^2/2$ being the Rayleigh distance. From Equation (5) the resulting propagated CSD turns out to be

$$
\begin{aligned}
W(\boldsymbol{r}_1, \boldsymbol{r}_2, z) \ = \ & \frac{2}{\pi\, w_z^2}\, e^{-(r_1^2 + r_2^2)/w_z^2}\, e^{-ik(r_1^2 - r_2^2)/(2R_z)} \times \\[2mm]
& \sum_{n,m} \lambda_{nm}\, \frac{n!}{(n+|m|)!} \left( \frac{2 r_1 r_2}{w_z^2} \right)^{|m|} L_n^{|m|}\!\left( \frac{2 r_1^2}{w_z^2} \right) L_n^{|m|}\!\left( \frac{2 r_2^2}{w_z^2} \right) e^{-im(\varphi_1 - \varphi_2)}
\end{aligned}
\tag{11}
$$

On comparing Equations (8) and (11) we see that, except for a spherical curvature term, the CSD maintains exactly the same form across any plane $z =$ constant, provided that $w_0$ is replaced by $w_z$. This is briefly expressed by saying that the partially coherent fields radiated by our sources are shape invariant. In particular, on letting $\boldsymbol{r}_1 = \boldsymbol{r}_2 = \boldsymbol{r}$ in Equation (11), we obtain the intensity

$$
I(\boldsymbol{r}, z) = \frac{2}{\pi\, w_z^2}\, e^{-2r^2/w_z^2} \sum_{n,m} \lambda_{nm}\, \frac{n!}{(n+|m|)!} \left( \frac{2r^2}{w_z^2} \right)^{|m|} \left[ L_n^{|m|}\!\left( \frac{2r^2}{w_z^2} \right) \right]^2 ,
\tag{12}
$$

which, besides maintaining the same shape under propagation, presents rotational invariance.

By virtue of the above property we can refer most of our considerations to the plane $z = 0$, omit the explicit dependence on $z$ of the functions and, for brevity, use the symbol $w$ (without any subscript) for the spot size of the modes.

### 2.2. Some Closed-Form CSDs with LG Modes

The most celebrated example of shape-invariant partially coherent beams are, without any doubt, the *Gaussian Schell-model* (GSM) beams. They were introduced more that forty years ago [2–5] and since then they have represented in a countless number of cases the archetype of partially coherent beams. This is mainly due to the simple mathematical form of their CSD.

The shape invariance of such beams can be traced back, of course, to the mathematical form of their coherent modes, which are, in fact, Hermite–Gaussian [10,11] or, equivalently, LG functions [25]. On putting eigenvalues of the form (see, for example, [26])

$$
\lambda_{nm} = \lambda_0\, \beta^{2n+|m|} \quad (0 \le \beta \le 1) ,
\tag{13}
$$

with $\lambda_0$ and $\beta$ positive parameters, into Equation (3), with the modes given in Equation (7), the following CSD is obtained:

$$
W(\boldsymbol{r}_1, \boldsymbol{r}_2) = A\, e^{-a(r_1^2 + r_2^2)}\, e^{-b(\boldsymbol{r}_1 - \boldsymbol{r}_2)^2} ,
\tag{14}
$$

where $A$ is a proportionality factor having dimensions of an intensity, while

$$
a = \frac{1}{w^2} \left( \frac{1-\beta}{1+\beta} \right) ; \quad b = \frac{2}{w^2} \left( \frac{\beta}{1-\beta^2} \right)
\tag{15}
$$

are two parameters related to the widths of the intensity profile and of the degree of coherence, respectively, which are

$$
I(\boldsymbol{r}) = A\, e^{-2ar^2} ; \quad \gamma(\boldsymbol{r}_1, \boldsymbol{r}_2) = e^{-b(\boldsymbol{r}_1 - \boldsymbol{r}_2)^2} .
\tag{16}
$$

Therefore, both the intensity and the degree of coherence of such sources are described by Gaussian functions. Furthermore, the degree of coherence is shift-invariant across the transverse section of the beam, that is, it only depends on the difference of the position vectors of the two considered points. Such a characteristic identifies the so-called *Schell-model sources*. It is interesting to note that the coherence area across the source (proportional to $1/b$) reduces on increasing $\beta$, i.e., on increasing the number of modes that significantly

contribute to the source [1]. This is a rather general result of the modal theory of coherence, although not universal (see Section 3).

An important generalization of the class of GSM source, due to Simon and Mukunda [26–28], is represented by the *Twisted Gaussian Schell-model* (TGSM) sources, whose CSD reads

$$W(\boldsymbol{r}_1, \boldsymbol{r}_2) = A \, e^{-a(r_1^2 + r_2^2)} \, e^{-b(\boldsymbol{r}_1 - \boldsymbol{r}_2)^2} \, e^{iu(\boldsymbol{r}_1 \times \boldsymbol{r}_2)_\perp} \, , \tag{17}$$

where $u$ is a real parameter called the *twist parameter* and the subscript $\perp$ denotes the $z$-component of a vector. Physical interpretations and possible realization schemes of these twisted sources can be found in Refs. [29,30]. The corresponding intensity profile and degree of coherence take the forms

$$I(\boldsymbol{r}) = A \, e^{-2ar^2} \, ; \quad \gamma(\boldsymbol{r}_1, \boldsymbol{r}_2) = e^{-b(\boldsymbol{r}_1 - \boldsymbol{r}_2)^2} \, e^{iu(\boldsymbol{r}_1 \times \boldsymbol{r}_2)_\perp} \, . \tag{18}$$

We can notice that the absolute value of $\gamma$ is still shift-invariant, but $\gamma$ itself is not.

Simon and Mukunda introduced such sources on the basis of symmetry arguments and also found their coherent modes and eigenvalues. Even in this case the modes are LG functions, with eigenvalues of the form

$$\lambda_{nm} = \lambda_0 \, \beta^{2n+|m|} \, \mu^{|m|/2} \quad (0 \leq \beta \leq 1; \, 0 \leq \mu \leq 1) \, , \tag{19}$$

and this makes such sources of the shape-invariant type. The relations between $(a, b)$ and $(w, \beta)$ are still given by Equation (15), while

$$u = 2b\left(\frac{\mu - 1}{\mu + 1}\right), \tag{20}$$

which implies

$$|u| \leq 2b \, , \tag{21}$$

that is, the maximum value of the twist parameter is bounded by the inverse of the (squared) width of the degree of coherence. Therefore, no twist can be present in a perfectly coherent source [26–30]. Research on partially coherent sources presenting a twist is still very active [31–42].

The next example we recall is that of the partially coherent *Flattened Gaussian* beams. They were introduced first in their coherent version [43], to provide a simple model to manage the paraxial propagation of light beams with a flattened profile. Their partially coherent counterpart was presented as an example of the superposition of mutually uncorrelated modes with helicoidal phases [44]. In that case, only a finite number of LG modes, all having $n = 0$, with equal eigenvalues was chosen. In particular, if

$$\lambda_{nm} = \lambda_0 \, \delta_{n,0} \times \begin{cases} 1 & (m = 0, 1, 2, \ldots, M) \\ 0 & (m > M) \end{cases} \tag{22}$$

with $\delta_{n,n'}$ the Kronecker symbol, the following CSD is obtained:

$$W_M(\boldsymbol{r}_1, \boldsymbol{r}_2) = \frac{A}{M!} \, e^{-(r_1^2 + r_2^2)/w^2} \exp\left[\frac{2r_1 r_2 \, e^{-i(\varphi_1 - \varphi_2)}}{w^2}\right] \Gamma_{M+1}\left[\frac{2r_1 r_2 \, e^{-i(\varphi_1 - \varphi_2)}}{w^2}\right] , \tag{23}$$

where $\Gamma_{M+1}(x)$ is the incomplete Gamma function [24]. The corresponding intensity reads

$$I_M(\boldsymbol{r}) = \frac{A}{M!} \, \Gamma_{M+1}\left(\frac{2r^2}{w^2}\right) . \tag{24}$$

Plots of the intensity and degree of coherence for such kind of sources are shown in [44]. While the former always presents a flat-topped circularly symmetric profile, interesting spatial structures can be observed for the degree of coherence.

To conclude this roundup of examples of closed-form CSDs with LG modes, we recall the one introduced by Ponomarenko in Ref. [45], which has been also recently experimentally synthesized [46]. It is obtained as the incoherent superposition of an infinite number of LG modes of different radial order, but with the same azimuthal index $m$. More precisely, the coefficients decrease geometrically with $n$, i.e.,

$$\lambda_{nm'} = \lambda_0 \, \delta_{m,m'} \, \xi^n \tag{25}$$

with $0 < \xi < 1$ and $m' = 0, \pm 1, \dots, \pm \infty$.

The resulting CSD is of the form

$$W_m(\boldsymbol{r}_1, \boldsymbol{r}_2) = A \, e^{-a(r_1^2 + r_2^2)} e^{-im(\varphi_1 - \varphi_2)} I_m(c \, r_1 r_2), \tag{26}$$

with

$$a = \frac{1}{w^2} \left( \frac{1 + \xi}{1 - \xi} \right); \quad c = \frac{1}{w^2} \left( \frac{4\sqrt{\xi}}{1 - \xi} \right) \tag{27}$$

and $I_m$ is the modified Bessel function of the first kind and order $m$ [24]. Note that in this case the modulus of the spectral degree of coherence is independent of the relative orientation of the points, i.e., it is circularly symmetric.

## 3. Laguerre-Christoffel-Darboux Sources

Here, we introduce a new type of shape-invariant CSDs whose expression can be given in closed form. We first limit ourselves to the case where the angular index ($m$) of the LG functions in the Mercer expansion in Equation (3) is kept fixed and take a finite number of modes with different $n$ and equal eigenvalues. In particular we take

$$\lambda_{nm'} = \lambda_0 \, \delta_{m,m'} \times \begin{cases} 1 & (n = 0, 1, 2, \dots, N) \, ; \\ 0 & (n > N) \, . \end{cases} \tag{28}$$

with $m' = 0, \pm 1, \dots, \pm \infty$.

Using the expression of the modes in Equation (7), the following expression for the resulting CSD turns out:

$$W_{Nm}(\boldsymbol{r}_1, \boldsymbol{r}_2) = A \, e^{-(r_1^2 + r_2^2)/w^2} e^{-im(\varphi_1 - \varphi_2)} \left( \frac{2r_1 r_2}{w^2} \right)^{|m|} G_{Nm}(r_1, r_2), \tag{29}$$

where

$$G_{Nm}(r_1, r_2) = \sum_{n=0}^{N} \frac{n!}{(n + |m|)!} \, L_n^{|m|} \left( \frac{2r_1^2}{w^2} \right) L_n^{|m|} \left( \frac{2r_2^2}{w^2} \right). \tag{30}$$

Note that the case $N = 0$ corresponds to a single Laguerre–Gaussian mode, so that the source is perfectly coherent.

The effect of taking the above coefficients is that the sum in Equation (30) can be calculated exactly on exploiting the Christoffel-Darboux formula [24,47]. The latter holds for any system of polynomials $\{f_n(x)\}$ on the (finite or infinite) interval $(a, b)$, which are orthonormal with respect to the non-negative weight function $w(x)$. The theorem states that, if $k_n$ is the coefficient of $x^n$ in the polynomial and

$$h_n = \int_a^b f_n^2(x) w(x) \, \mathrm{d}x, \tag{31}$$

then

$$\sum_{n=0}^{N} \frac{f_n(x)f_n(y)}{h_n} = \frac{k_N}{k_{N+1}h_N} \frac{f_{N+1}(x)f_N(y) - f_N(x)f_{N+1}(y)}{x - y} . \tag{32}$$

Using the above result, Equation (30) can be written as

$$G_{Nm}(r_1, r_2) = \frac{(N+1)!}{(N+|m|)!} \frac{L_N^{|m|}\left(\frac{2r_1^2}{w^2}\right)L_{N+1}^{|m|}\left(\frac{2r_2^2}{w^2}\right) - L_{N+1}^{|m|}\left(\frac{2r_1^2}{w^2}\right)L_N^{|m|}\left(\frac{2r_2^2}{w^2}\right)}{\frac{2}{w^2}(r_1^2 - r_2^2)} \tag{33}$$

if $r_1 \neq r_2$. In the limit of $r_1 = r_2 = r$, the resulting indeterminate can be evaluated by means of the de l'Hôpital theorem [24], which gives

$$G_{Nm}(r, r) = \frac{(N+1)!}{(N+|m|)!}\left[L_N^{|m|}\left(\frac{2r^2}{w^2}\right)L_N^{|m|+1}\left(\frac{2r^2}{w^2}\right) - L_{N+1}^{|m|}\left(\frac{2r^2}{w^2}\right)L_{N-1}^{|m|+1}\left(\frac{2r^2}{w^2}\right)\right]. \tag{34}$$

A source characterized by the CSD in Equation (29), with the function $G_{Nm}$ given in Equation (33) will be called a *Laguerre-Christoffel-Darboux* (LCD) source. Sources of this class could be experimentally generated following the procedures described in Refs. [41,46].

From Equations (29), with $r_1 = r_2 = r$, and (34) we obtain the following expression for the intensity:

$$I_{Nm}(\mathbf{r}) = \frac{2}{\pi w^2} e^{-2r^2/w^2}\left(\frac{2r^2}{w^2}\right)^{|m|} \frac{(N+1)!}{(N+|m|)!} \times$$

$$\left[L_N^{|m|}\left(\frac{2r^2}{w^2}\right)L_N^{|m|+1}\left(\frac{2r^2}{w^2}\right) - L_{N+1}^{|m|}\left(\frac{2r^2}{w^2}\right)L_{N-1}^{|m|+1}\left(\frac{2r^2}{w^2}\right)\right], \tag{35}$$

whose behavior is shown in Figures 1 and 2 for some values of $N$ and $m$. Figure 1 shows the changes in the intensity profile when $m$ is kept fixed and the number of modes is varied. It can be noticed that, due to the presence of the optical vortex, the intensity profile shows a central dark region where the intensity is practically zero. The radius of this circular dark region increases with $m$ and decreases with $N$. We note that the presence of a central dark zone with a fast variation of the intensity at the border of this zone could be efficiently used in particle trapping. Furthermore, $N + 1$ relative maxima are observed in the axial profile, which become less and less evident on increasing the value of the indices. Some intensity profiles across the source plane are shown in Figure 2. Due to the shape invariance of the LCD sources, Figures 1 and 2 are valid at any transverse plane, the only change being the scale factor $w$ that varies with $z$ [see Equation (10)]. The same holds for the forthcoming figures of intensity and degree of coherence.

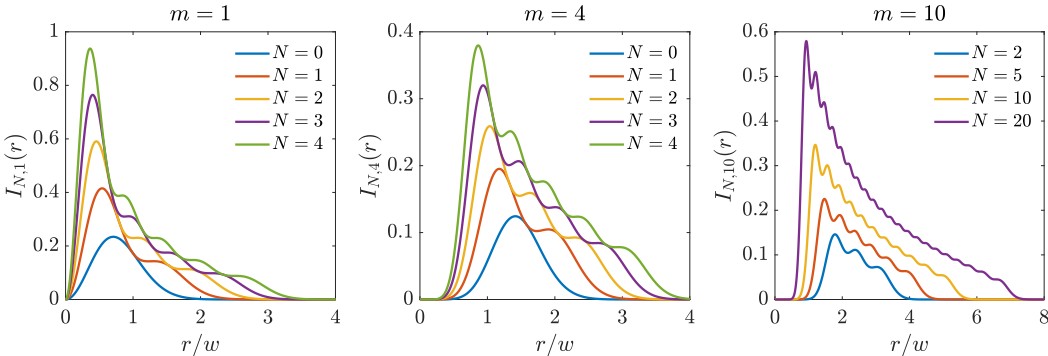

**Figure 1.** Intensity profiles $I_{Nm}$ given by Equation (35) for several values of $N$ and $m$ (as indicated in Figure labels).

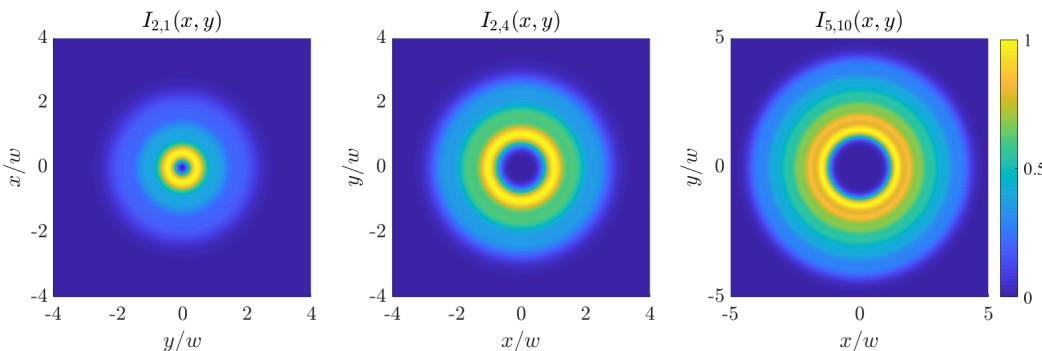

**Figure 2.** Maps of the intensity $I_{Nm}$ (normalized to the maximum) across the source plane for three different pairs $(N, m)$ (as indicated in the figure labels).

Interesting results are obtained on considering the limit of the expression in Equation (35) for large values of $N$. Using Equation (8.978.2) of Ref. [48], that is,

$$\lim_{N \to \infty} \frac{1}{N^m} L_N^m \left( \frac{\xi}{N} \right) = \xi^{-m/2} J_m(2\sqrt{\xi}) , \tag{36}$$

where $J_m$ is the Bessel function of the first kind and order $m$ [24], the following asymptotic expression is obtained for the source intensity:

$$I_{Nm}(\boldsymbol{r}) \simeq \frac{\sqrt{2}}{\pi w} e^{-2r^2/w^2} \frac{(N+1)!}{(N+|m|)!} \frac{1}{r} \left[ N^{|m|+1/2} J_{|m|} \left( \frac{2r\sqrt{2N}}{w} \right) J_{|m|+1} \left( \frac{2r\sqrt{2N}}{w} \right) \right. \\ \left. - \sqrt{N-1}(N^2-1)^{|m|/2} J_{|m|} \left( \frac{2r\sqrt{2(N+1)}}{w} \right) J_{|m|+1} \left( \frac{2r\sqrt{2(N-1)}}{w} \right) \right]. \tag{37}$$

The usefulness of introducing the Bessel functions is apparent if we consider their property according to which $J_k(x)$ is negligibly small for $x < k$ [24,49]. This allows us to roughly estimate the radius of the central dark region, say $r_0$. In fact, if we neglect the difference between $m$ and $m+1$, as well as that between $N$ and $N+1$, we can state that all the Bessel functions appearing in Equation (37) vanishes if

$$\frac{2r\sqrt{2N}}{w} \lesssim |m| , \tag{38}$$

so that

$$r_0 \approx \frac{|m|}{2\sqrt{2N}} w . \tag{39}$$

Radial profiles and 2D plots of the intensity profile are shown in Figure 3 for $N = 100$ and $m = 25, 50$, and $100$, corresponding to $r_0$ values of 0.9 w, 1.8 w, and 3.5 w, respectively. It is worth noting that the intensity drops practically to zero outside the donut region. For example, the intensity $I_{100,25}(r)$ is below $10^{-4}$ of its maximum for $r < 0.6w$ or $r > 15.5$ w. The same happens for $I_{100,50}(r)$ if $r < 1.3$ w or $r > 16.3$ w, and for $I_{100,100}(r)$ if $r < 2.6$ w or $r > 17.5$ w. The inset in Figure 3 shows the ripples of the intensity profile, that become smoother and smoother on increasing $N$.

As far as the complex degree of coherence is concerned, we have

$$\gamma_{Nm}(\boldsymbol{r}_1, \boldsymbol{r}_2) = \frac{G_{Nm}(r_1, r_2) \, e^{im(\varphi_2 - \varphi_1)}}{\sqrt{G_{Nm}(r_1, r_1) G_{Nm}(r_2, r_2)}} . \tag{40}$$

It is seen that the absolute value of $\gamma_{Nm}$ depends only on the radial distances of the considered points from the source center, so that the source exhibits perfect coherence along any annulus that is concentric to the source center, i.e.,

$$|\gamma_{Nm}(r, \varphi_1; r, \varphi_2)| = 1 \tag{41}$$

for any choice of $m$, $\varphi_1$ and $\varphi_2$. In this sense, the sources present circular coherence [49,50]. Conversely, coherence can be partial or even vanishing between two points at different distances from the center along a radius.

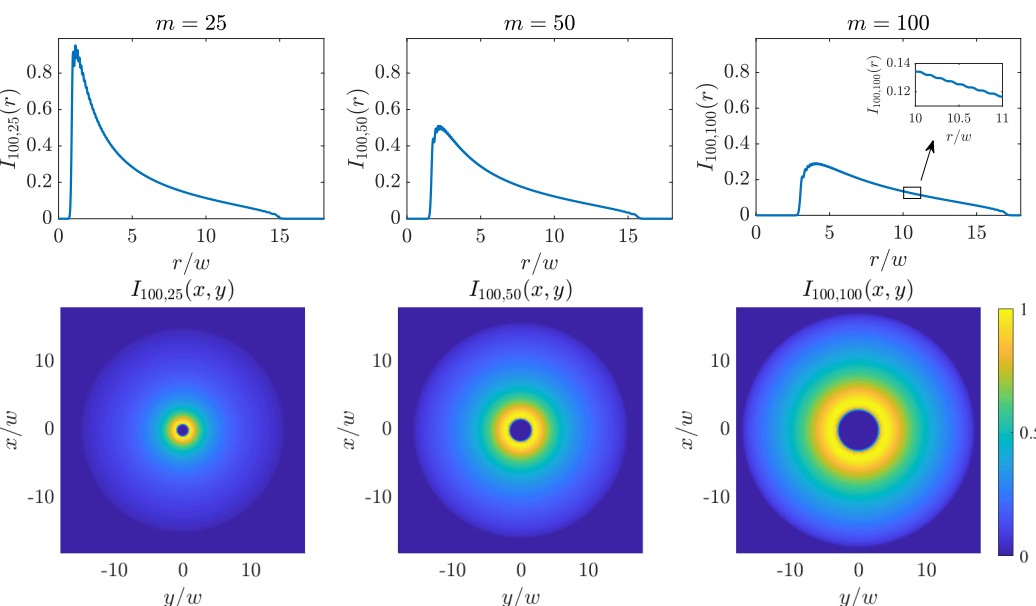

**Figure 3.** Intensity profiles as functions of the radial coordinate and intensity maps (normalized to its maximum) for $N = 100$ and $m = 25, 50, 100$.

Plots of the absolute value of the degree of coherence across the plane $(r_1, r_2)$ are shown in Figure 4. It can be observed that setting $r_2$ $(r_1)$ close to zero and varying $r_1$ $(r_2)$, it passes through $N$ zeros. The same happens for large values of $r_2$ $(r_1)$, i.e., at points outside the intensity donut. The positions of such minima along $r_1$ $(r_2)$ remain almost the same on varying $r_2$ $(r_1)$ and are located within the intensity donut (for $m \neq 0$).

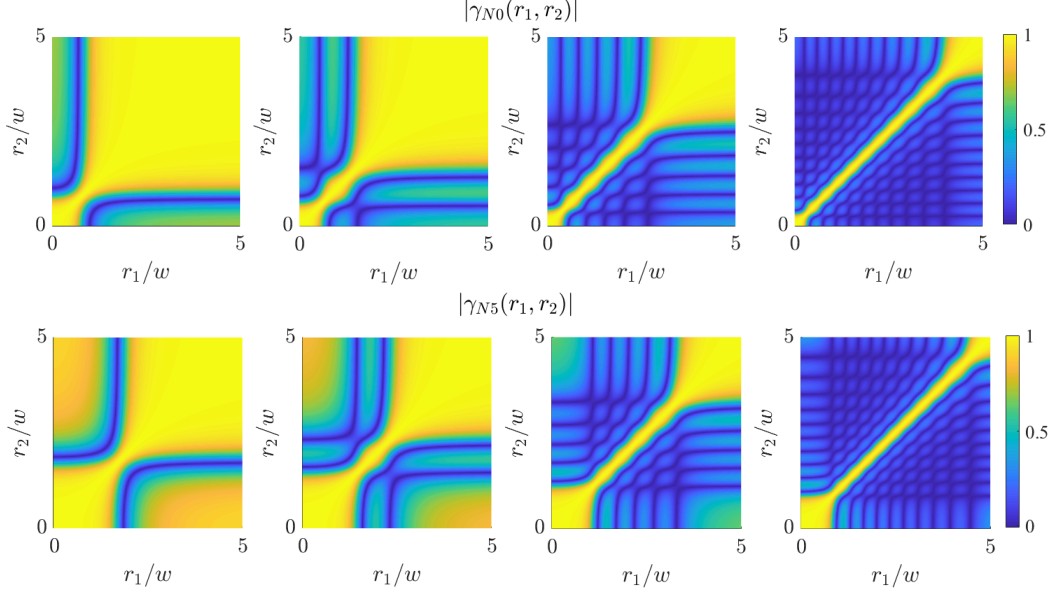

**Figure 4.** Absolute value of the complex degree of coherence and funtion in the $r_1$ $r_2$ plane for $N = 1, 2, 5, 10$ (from left to right) and $m = 0$ (upper row) and $m = 5$ (lower row).

Maps of the degree of coherence across the source plane can be seen in Figure 5, which shows the absolute value of $\gamma_{Nm}(\mathbf{r}_1, \mathbf{r}_2)$ as a function of $\mathbf{r}_1$ for three different values of $\mathbf{r}_2$. The three positions $\mathbf{r}_2$ are denoted by small dots. Chosen parameters are $N = 2$ and $m = 4$, for which the intensity map is the one shown in the center part of Figure 2.

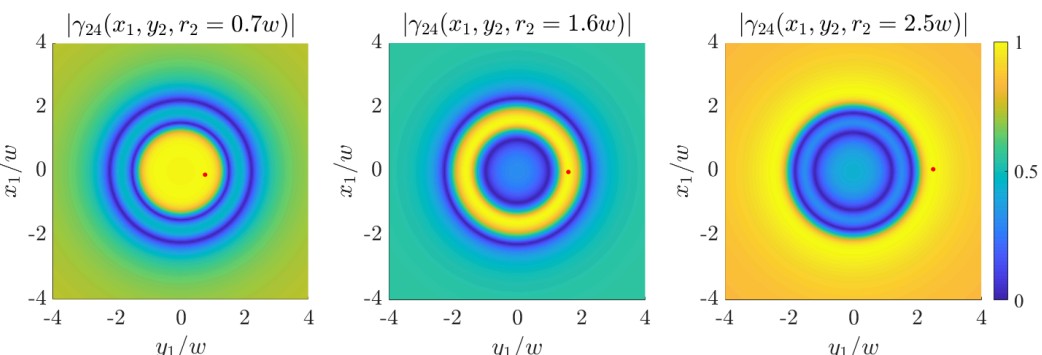

**Figure 5.** Absolute degree of coherence (Equation (40)) relative to three different points, indicated by red dots, for $N = 2$, $m = 4$.

Asymptotic expressions of the degree of coherence can be also deduced from Equations (33) and (40). In fact, keeping $r_2$ fixed and increasing $r_1$ we obtain

$$\lim_{r_1 \to \infty} |\gamma_{Nm}(r_1, \varphi_1; r_2, \varphi_2)| =$$

$$\frac{\dfrac{1}{\sqrt{N+1}} \left| L_N^{|m|}\left(\dfrac{2r_2^2}{w^2}\right) \right|}{\sqrt{L_N^{|m|}\left(\dfrac{2r_2^2}{w^2}\right) L_N^{|m|+1}\left(\dfrac{2r_2^2}{w^2}\right) - L_{N+1}^{|m|}\left(\dfrac{2r_2^2}{w^2}\right) L_{N-1}^{|m|+1}\left(\dfrac{2r_2^2}{w^2}\right)}}. \tag{42}$$

In particular, if $r_2$ is close to zero,

$$\lim_{\substack{r_1 \to \infty \\ r_2 \to 0}} |\gamma_{Nm}(r_1, \varphi_1; r_2, \varphi_2)| = \lim_{\substack{r_2 \to \infty \\ r_1 \to 0}} |\gamma_{Nm}(r_1, \varphi_1; r_2, \varphi_2)| = \sqrt{\frac{|m|+1}{N+|m|+1}} \tag{43}$$

The absolute value of the degree of coherence along a radius is also represented in Figure 6 as a function of $r_1$ for three different values of $r_2$. The order $N$ is fixed (equal to 5) and several values of $m$ are considered ($m = 5, 10, 20$). The three values of $r_2$ are chosen as follows: the first one corresponds to a point close to the inner border of the intensity donut; the second one is in the central part of the donut; and the third one is near the outer border of the donut. For any of these sources, complete correlation is observed for any pair of points located on the same circle concentric with the source center, as it was expected from Equation (41). Furthermore, within the region where the intensity is not negligible, the absolute value of the degree of coherence shows $N$ minima, where it vanishes, meaning that there could be complete incoherence between points belonging to different circles. Outside that region the absolute degree of coherence tends to a limit given by Equation (42).

Similar plots are shown in Figure 7 for a source where a larger number of modes are considered ($N = 20$) with topological charge $m = 0, 1, 2$. For these cases, it can be observed that the degree of coherence is considerably high for pairs of points in the region where the intensity is significant. For example, for $m = 0$ the degree of coherence is over 0.75 for any pair of points located within a circle of radius $0.2w$ where the intensity is over 0.2 of its maximum value. That is, a relatively highly coherent source is obtained, despite we are considering the incoherent superposition of a high number of modes. A similar behavior is

observed for topological charge $m = 1$ and $m = 2$ and a ring around the circle where the maximum intensity is reached.

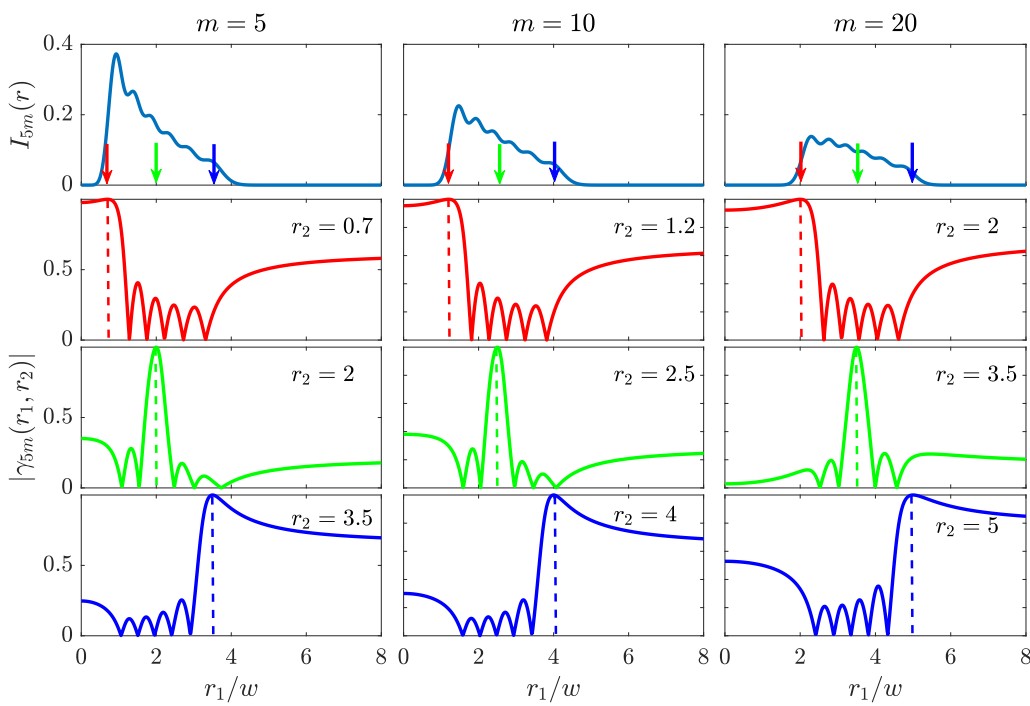

**Figure 6.** Upper row: intensity profile given by Equation (35) for $N = 5$ and several values of $m$. Second, third and fourth rows: absolute value of the degree of coherence for relative to a point located at a distance $r_2$ from the source center (indicated with an arrow on the intensity profile and a dashed line in the degree of coherence profile).

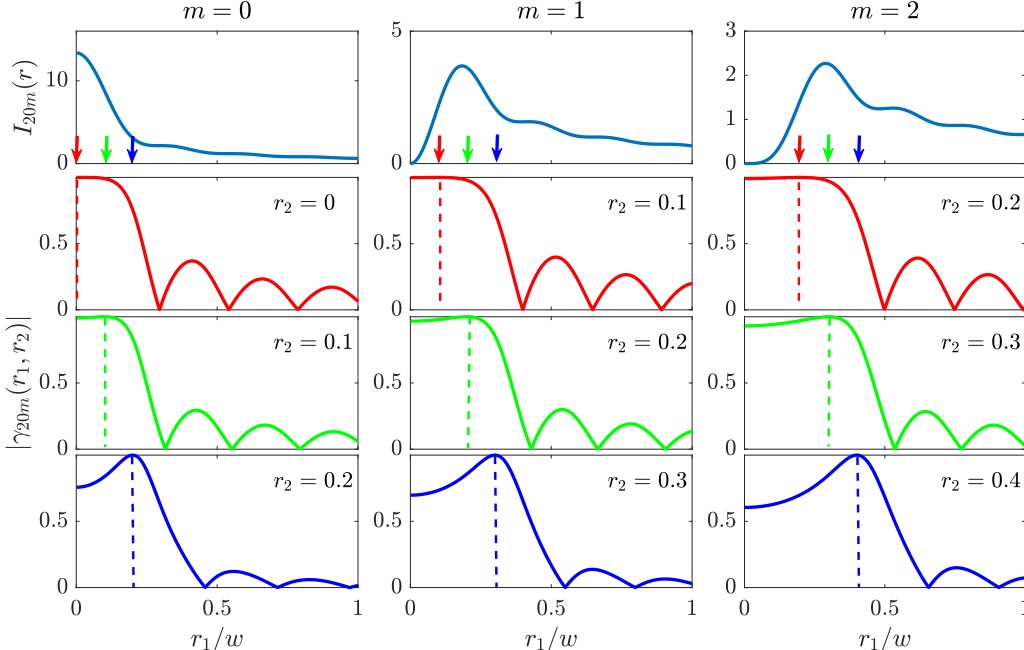

**Figure 7.** Upper row: intensity profile given by Equation (35) for $N = 20$ and several values of $m$. Second, third and fourth rows: absolute value of the degree of coherence for relative to a point located at a distance $r_2$ from the source center (indicated with an arrow on the intensity profile and a dashed line in the degree of coherence profile).

A source with a more structured degree of coherence is obtained if further terms are included into the Mercer expansion. In particular, if we take the modes of orders both $(n, m)$ and $(n, -m)$, with the same weight, i.e.,

$$\lambda_{nm'} = \lambda_0 \frac{\delta_{m',m} + \delta_{m',-m}}{2} \times \begin{cases} 1 & (n = 0, 1, 2, \ldots, N) \\ 0 & (n > N) \end{cases} \tag{44}$$

the intensity profile does not change with respect to the previous case, but an angular modulation is introduced into the degree of coherence, being

$$\gamma_{Nm}(\mathbf{r}_1, \mathbf{r}_2) = \frac{G_{Nm}(r_1, r_2) \, \cos[m(\varphi_2 - \varphi_1)]}{\sqrt{G_{Nm}(r_1, r_1) G_{Nm}(r_2, r_2)}} \, . \tag{45}$$

Figures 8 and 9 show the behavior of the degree of coherence for several sources of this type. In this case the absolute value of the degree of coherence does not only depend on the radial coordinates but also on the angle difference of the considered points, except when $m = 0$. In fact, if $m \neq 0$ the degree of coherence is zero for points that satisfy $\varphi_2 - \varphi_1 = \pi(\ell + 1/2)/m$, while is unitary for points with $\varphi_1 - \varphi_2 = \pi\ell/m$. For other angles, the degree of coherence is modulated by a sinusoidal function.

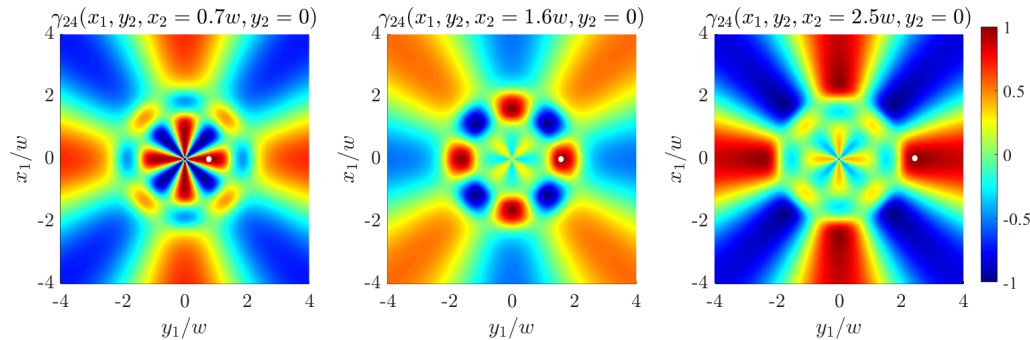

**Figure 8.** Degree of coherence for a source described by Equations (44) and (45), $N = 2$, $m = 4$, relative to three different points indicated by white dots and in the labels.

Note that sources described by Equation (8) with the eigenvalues given by Equation (28) or Equation (44), with odd $m$, present a degree of coherence that reaches the value $-1$ for any pair of diametrically opposite points on the same concentric circle (see Figure 9). Partially coherent beams with analogous characteristics have been recently considered, both theoretically and experimentally, to overcome the classical Rayleigh diffraction limit in the imaging of two holes (see for example [51–54]).

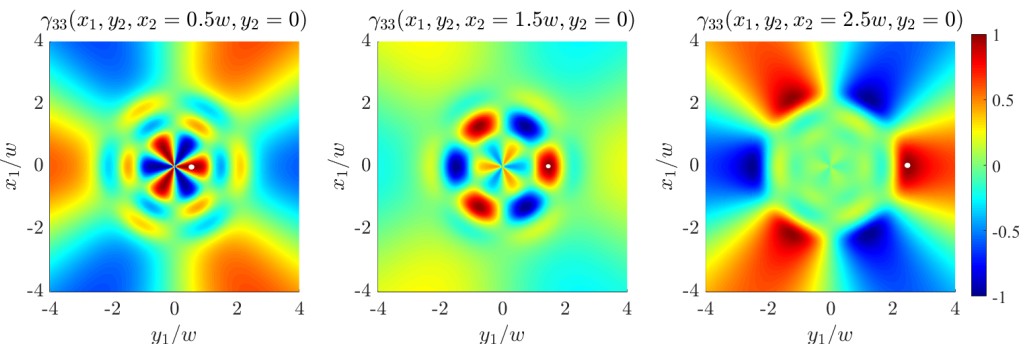

**Figure 9.** Degree of coherence for a source described by Equations (44) and (45), $N = 3$, $m = 3$, relative to three different points indicated by white dots and in the labels.

## 4. Conclusions

In this paper we have introduced a new class of partially coherent fields whose cross-spectral density is shape invariant upon paraxial propagation. This means that both their transverse intensity profile and the structure of their degree of spectral coherence keep their form unchanged during propagation, except for a transverse scaling factor.

Fields of this class are obtained on superimposing a finite number of mutually uncorrelated Laguerre–Gaussian modes, having fixed topological charge and equal power, but the resulting CSD can be always expressed in a simple mathematical form regardless the number of involved modes. The closed-form expression of the CSD can be exploited for deducing properties of the beams analytically, or for making the problem of their propagation easier when a numerical approach has to be used (e.g., in non-ABCD or non-deterministic systems).

The intensities of the beams we have considered have rotational symmetry and (for non vanishing topological charge of the component modes) have a donut-like profile, whose size can be varied at will on changing the parameters of the modes. An interesting feature of these profiles is that the intensity is uniform and practically zero in the whole regions inside the donut, thus mimicking a circular-well potential in particle trapping experiments.

The degree of coherence presents a vortex-phase structure and its modulus only depends on the distances of the involved points from the propagation axis. It is interesting to remark that, contrary to what usually happens, for low values of the topological charge (including the case of vanishing charge), the coherence area across the transverse plane remains relatively large in whole region where the intensity is significant, even if a very large number of component modes is considered.

The vortex structure of the degree of coherence can be converted into an angularly periodic modulation on superimposing modes with opposite charge, leaving the radial dependence unchanged. For odd values of the charge, the degree of coherence equals $-1$ for any pairs of diametrically opposite points on a circle $r =$ constant.

Sources with the above characteristics could have potential applications in several fields, such as particle trapping, optical communications, imaging, and more.

**Author Contributions:** All authors contributed equally. All authors have read and agreed to the published version of the manuscript.

**Funding:** This research was funded by Ministerio de Economía y Competitividad, grant number PID2019-104268GB-C21.

**Institutional Review Board Statement:** Not applicable.

**Informed Consent Statement:** Not applicable.

**Data Availability Statement:** Not applicable.

**Acknowledgments:** The authors wish to thank F. Gori for his useful suggestions.

**Conflicts of Interest:** The authors declare no conflict of interest. The funders had no role in the design of the study; in the collection, analyses, or interpretation of data; in the writing of the manuscript, or in the decision to publish the results.

## Abbreviations

The following abbreviations are used in this manuscript:

| | |
|---|---|
| CSD | Cross-spectral density |
| GSM | Gaussian Schell-model |
| LG | Laguerre–Gaussian |
| LCD | Laguerre–Christoffel–Darboux |
| OAM | Orbital angular momentum |
| TGSM | Twisted Gaussian Schell-model |

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
