# Peer review of "A New Type of Shape-Invariant Beams with Structured Coherence: Laguerre-Christoffel-Darboux Beams"

_photonics, doi:10.3390/photonics8040134_

Round 1

Reviewer 1 Report

Martínez-Herrero et al. in this work introduced a new class of partially coherent fields whose cross-spectral density (including the intensity and degree of coherence) is shape invariant upon paraxial propagation. The derived results are interesting and solid. The manuscript is well written. Hence, I definitely recommend it to be accepted by your journal. However, I have only one suggestion that the authors should give the suggestions for the experimental realization of such beams in the manuscript. The similar experiments are introduced in many works, such as Refs. 39 and 44.

Reviewer 2 Report

In this paper, the authors investigated the fields whose cross-spectral density keeps shape during paraxial propagation. A finite number of mutually uncorrelated Laguerre–Gaussian modes are superimposed to obtain the fields. Besides, intensity distributions and the degree of coherence can be controlled by N and m, namely, the number of the modes and topological charge.

The expressions of this paper are clear and coherent. The research is innovative, so I recommend the manuscript to be published after explaining some questions as follows:

  • In figure2, why not select I(5, 4)?
  • The propagation dynamics with a pair (N, m) on x(y)-z plane should be shown and discussed.
  • The possible experimental realization techniques should be clarified.
  • What are the potential applications of such beams?
  • The sentence in line123, page7 is not clear.

Reviewer 3 Report

The authors analyze the coherence properties of beams due to an incoherent superpositon of Laguerre-Gauss modes. This concerns in particular, intensity , shape and field correlation at two distinct points of the plane. This is a problem of interest due to potential applications in free space communications and others like particle trapping. The formalism of LCD partially coherent fields is correctly developed and figures illustrate the evolution of the intensity profiles and degrees of coherence of the beams. I outline the following questions to consider in the final version of the paper : 

-Include a Fig  to introduce the beam propagation along z direction, the radial coordinates, and the selected point positions at r1 - r2                          -It is recommended to develop the physical signification of the shape invariant LCD source and its degree of coherence which exhibit maximum and zero values in Figs 6-7. This will be clearly explained to support the formula issued from the model.                                                                        -It is also of interest to discuss which parameters may  disturb the shape invariance, in particular contributions of beam aberrations and others wave perturbations along the z propagation.                                                          -- -The capability of overcoming the Rayleigh limit is noted, this point merit to  include few comments with  the LG - LCD mode beams.   

To conclude, the manuscript develops the fairly complete analysis and modeling of these novel beams as well as their degree of coherence. It can be accepted in the journal but it is required to take account of the questions and comments of the review in the final form of the paper.   

Reviewer 4 Report

See attachment
